# MicroRNAs: Small but Key Players in Viral Infections and Immune Responses to Viral Pathogens

**DOI:** 10.3390/biology12101334

**Published:** 2023-10-14

**Authors:** Anais N. Bauer, Niska Majumdar, Frank Williams, Smit Rajput, Lok R. Pokhrel, Paul P. Cook, Shaw M. Akula

**Affiliations:** 1Department of Microbiology & Immunology, Brody School of Medicine, East Carolina University, Greenville, NC 27834, USA; bauera22@students.ecu.edu (A.N.B.); majumdarn22@students.ecu.edu (N.M.); williamsjohnf@ecu.edu (F.W.); 2Department of Internal Medicine, Brody School of Medicine, East Carolina University, Greenville, NC 27834, USA; rajputs20@ecu.edu; 3Department of Public Health, Brody School of Medicine, East Carolina University, Greenville, NC 27834, USA; pokhrell18@ecu.edu

**Keywords:** microRNA, miRNA, viral infection, miRNA therapeutics, miRNA diagnostics

## Abstract

**Simple Summary:**

Viral outbreaks continue to be an obstacle to human health, with the Severe Acute Respiratory Syndrome-related Coronavirus (SARS-CoV-2) pandemic shedding light on the current vulnerabilities of our healthcare system. Understanding viral pathology is paramount in developing methods to combat infection. The study of microRNAs (miRNAs) is relatively new to the world of virology research. These small strands of nucleotides are a versatile tool in the regulation of gene expression. Various miRNAs have roles in immune development, immune and inflammatory responses, and viral infections. Changes in miRNA expression can be a double-edged sword, used to alter cell activities to help the host fight infections or taken advantage of by viruses to enhance infection. Uncovering these interactions and their implications can provide direction for therapeutic and diagnostic advancement. Further, miRNAs may have the potential to predict the severity of viral infection and possible health outcomes, as well as track disease progression to inform treatment options. miRNA technology is also anticipated to be highly marketable and has already entered the realm of commercial biopharmaceuticals. Continuing to elucidate the functions of miRNA during infection and the therapeutic potential of these molecules will contribute to new strategies in the battle against current and future viral pathogens.

**Abstract:**

Since the discovery of microRNAs (miRNAs) in C. elegans in 1993, the field of miRNA research has grown steeply. These single-stranded non-coding RNA molecules canonically work at the post-transcriptional phase to regulate protein expression. miRNAs are known to regulate viral infection and the ensuing host immune response. Evolving research suggests miRNAs are assets in the discovery and investigation of therapeutics and diagnostics. In this review, we succinctly summarize the latest findings in (i) mechanisms underpinning miRNA regulation of viral infection, (ii) miRNA regulation of host immune response to viral pathogens, (iii) miRNA-based diagnostics and therapeutics targeting viral pathogens and challenges, and (iv) miRNA patents and the market landscape. Our findings show the differential expression of miRNA may serve as a prognostic biomarker for viral infections in regard to predicting the severity or adverse health effects associated with viral diseases. While there is huge market potential for miRNA technology, the novel approach of using miRNA mimics to enhance antiviral activity or antagonists to inhibit pro-viral miRNAs has been an ongoing research endeavor. Significant hurdles remain in terms of miRNA delivery, stability, efficacy, safety/tolerability, and specificity. Addressing these challenges may pave a path for harnessing the full potential of miRNAs in modern medicine.

## 1. Introduction

The central dogma of biology posits a deceivingly straightforward process. It describes how DNA is transcribed into RNA, which is then translated into protein, or not. In fact, only 1–2% of the transcribed genome encodes proteins in mammals [1]. Most of the DNA in eukaryotic cells is non-coding. 

MicroRNA (miRNA) is one of the many types of non-coding RNA found in eukaryotes [2]. Despite their average length of a mere ~22 base pairs, these small strands of nucleotides are predicted to regulate more than 60% of all human protein-coding genes [3]. They typically work at the post-transcriptional level to enhance or repress the translation of mRNA, though some evidence suggests they may also have regulatory functions at the transcriptional level [4]. But how does something so small make such a big impact? The minute size of miRNA contributes to its ability as a strong regulatory element. Functionally, the short sequence may allow room for multiple miRNAs to bind to a single messenger RNA (mRNA) and synergistically repress translation [5]. On the other hand, the target sequence in mRNA is also short and may be common between different groups of mRNAs [6]. Thus, a single miRNA has the potential to bind to many types of mRNAs. The regulatory effect of miRNA is further amplified by its ability to be released for reuse after its target has been degraded [4]. These traits allow miRNAs to act alone or in combination to rapidly suppress gene expression and protein production. 

Because of its key role in regulation, the miRNA biogenesis pathway is highly conserved between species [7]. It begins in the nucleus, where the corresponding gene is transcribed into primary miRNA (pri-miRNA) [7]. This pri-miRNA is then modified, and the resulting precursor miRNA (pre-miRNA) is transported to the cytoplasm via nuclear pores [6,7]. Once in the cytosol, precursor miRNA is processed, and the mature miRNA duplex binds to the RNA-induced silencing complex (RISC) [6]. The Argonaute (AGO) protein family within RISC, along with various cofactors, unwind the miRNA duplex and select one of the strands [6]. The AGO protein is largely responsible for transporting the complex to a mRNA with a short complimentary sequence [7]. In animals, this sequence is often in the 3′ untranslated region (UTR) of the mRNA with a perfect or near-perfect ~7-nucleotide base pairing with the seed sequence in miRNA [8]. This interaction typically triggers the mRNA degradation pathway to downregulate protein production [9]. In some instances, miRNA can alternatively interact with the 5′ UTR or coding sequence (CDS) of mRNA to affect stability or translation [10,11,12,13,14]. 

Translation may be directly repressed by miRNA through decreased ribosomal interaction with the mRNA [9]. More often, the target mRNA is destabilized, accelerating the standard degradation process [9]. In rare cases where the base pairing is much more extensive, AGO may directly cleave mRNA, making it susceptible to rapid degradation [4,9]. In contrast, miRNA has been shown to upregulate translation in some cases [15,16]. It may also directly regulate transcription via promotor interactions or indirectly by targeting the mRNA of transcriptional regulators [4,17]. Additionally, miRNA can bind to non-coding RNAs, including long non-coding RNA (lnc-RNA), circular RNA (circ-RNA), and other miRNAs [18].

The role of miRNAs in human health continues to be an area of active research. Since the discovery of miRNA in *C. elegans* in 1993 and mammalian miRNA in 2000, the field of miRNA research has grown dramatically [7]. In 2017 alone, there were ~11,000 studies on miRNA, with more than 6000 of these studies on miRNA diagnostics and therapeutics combined [19]. There is evidence of miRNA involvement in many medical conditions, including cardiovascular disease [20], various cancers [18,21,22,23,24,25], and infections [10,21,22,23,24,25]. This review is aimed at succinctly summarizing the latest findings in (i) mechanisms underpinning miRNA regulation of viral infection and (ii) the host immune response to viral pathogens, (iii) miRNA diagnostics and therapeutics targeting viral pathogens, and (iv) the miRNA market landscape.

## 2. Modes of miRNA-Mediated Gene Regulation 

Gene regulation by miRNA-RISC may occur through one of the following mechanisms: (i) site-specific cleavage, (ii) mRNA degradation, and (iii) translation inhibition [26,27]. Site-specific cleavage occurs when there is a near-perfect match of miRNA to the target RNA. This process is called RNA interference (RNAi). In contrast, miRNA-directed mRNA degradation and translation inhibition are the non-cleavage modes of miRNA regulation. As early as 2004, it was demonstrated by the Bartel lab that miR-196 can effectively regulate gene expression by directed cleavage of HOXB8 mRNA [28]. Such miRNA cleavage activity utilizes AGO proteins [29]. miRNA-directed endonucleolytic cleavage of target mRNAs is more common in plants than animals but has a critical function [30,31]. In animals, cleavage of miRNA targets is a poorly understood concept, partly due to a lack of effort to investigate this phenomenon. Recent studies have demonstrated several animal miRNAs to target mRNA by cleavage, and a few examples of such miRNAs are Let-7 (targeting TUSC2) [32], miR-92 (targeting SERBP1) [33], miR-127 and miR-136 (targeting Rtl1/Peg11) [34].

Multiple mechanisms are in place to regulate the miRNA-directed non-cleavage mode of regulating mRNA expression. The key event for all these activities is interactions between the scaffold protein, GW182, and any of the AGO proteins. The most common pathway is GW182-mediated deadenylation, followed by de-capping and mRNA degradation via NOT/CCR4/CAF1 complexes [35]. Other pathways include GW182 competing with eIF4G in association with poly-A binding protein (PABP) to prevent the circularization required for efficient translation of target mRNA [36,37]; GW182 preventing the formation of a functional 80S ribosome crucial to the translational process of the target transcript [27,38]; or GW182/AGO complex-induced ribosomal stalling that induces a translation elongation block [39]. Such diversity in miRNA-mediated silencing mechanisms provides life with enhanced capacity for gene regulation but also poses challenges to a complete understanding of the biology of miRNAs.

Apart from plant and animal miRNAs, there are also miRNAs encoded by viruses (v-miRs). More than 250 v-miRs have been reported to play crucial roles in virus pathogenesis [40]. Once again, perfect complementarity results in mRNA site-specific cleavage, as observed in simian virus 40 (SV40) encoded miR-S1-5p and miR-S1-3p, which target the mRNA coding for a protein known as Large T antigen [41]. In contrast, imperfect/partial complementarity with the target mRNA leads to translational repression via non-cleavage mode, as reported for EBV-encoded miR-BART-1p, miR-BART16, and miR-BART17-5p, which target latency-associated membrane protein LMP-1 [42]. Viral miRNAs, perhaps, have evolved cleavage or non-cleavage modes of regulating mRNA expressions based on necessity. For example, if only one miRNA is required to regulate a key mRNA for viral replication, it may follow the cleavage mode as in the case of SV40 miR-S1-5p [41]. If there are multiple v-miRs regulating one viral mRNA, as in the case of EBV-encoded miR-BART-1p, miR-BART16, and miR-BART17-5p, it may follow a non-cleavage mode [42]. Recent studies have also determined the ability of v-miRs to regulate cellular genes critical to their survival [14,43].

## 3. Host Response to Viral Pathogens

The ability to defend against harmful agents is an evolutionary necessity [44]. Even bacteria have developed a method to protect themselves against viral infection, using the clustered regularly interspaced palindromic repeat (CRISPR)/Cas system [45]. Humans, however, have a more complex strategy with many components. Elements of the immune system are distinguished historically as innate or adaptive [46]. Though classically defined as separate, recent research indicates crosstalk between these two branches [46]. In addition, it suggests communication between the immune system, nervous system, endocrine system, and microbiome [46]. 

Innate immunity is often referred to as the first line of defense and recognizes conserved antigens, including common motifs in pathogens [47]. It includes physical barriers, such as skin and the mucosal lining of the respiratory and digestive tracts [46]. At birth, infants are exposed to the external environment for the first time and must be able to survive interactions with other organisms, including viruses [48]. Human breast milk helps strengthen a newborn’s immune system and delivers ~1400 miRNAs, including many linked to immune system maturation and viral defenses [49].

When a novel pathogen breaches the physical barrier of the skin or mucous membranes, the innate immune response is non-specific and quick, mounting a defense in just minutes to hours [44]. The activation of adaptive immune cells occurs during the first exposure to a pathogen. When the body reencounters the same virus, its memory bank of immune cells developed during the first exposure quickly identifies the markers and responds efficiently [44]. While the innate immune response still occurs, the resident adaptive immune cells assemble more rapidly from memory to provide a strong wave of protection [44]. 

## 4. miRNA and Immunity

The impact of miRNAs on the immune system is observed before birth and has dynamic effects throughout the human lifetime [50,51]. For example, miR-181a has many targets involved in immune cell processes [51,52,53,54]. It is involved in T cell development, homeostasis, activation, and proliferation [51]. In addition, miR-181a-5p may play a role in B cell development from precursors in bone marrow, and miR-181 in Natural Killer (NK) cells promotes development from Hematopoietic Progenitor Cells (HPCs) [52,53]. miR-181a has been implicated in inhibiting the production of IL-1a and other inflammatory factors in macrophages and monocytes [54]. It also regulates monocyte-derived dendritic cell activation and the release of inflammatory cytokines [55]. This one type of miRNA has many roles in immune cell development, differentiation, expansion, activation, and effector functions. 

Several other miRNAs influence similar processes [50,56,57,58,59,60,61]. For example, the differential expression of miR-126, miR-146a, miR-150, and miR-17-92 is involved in the regulation of T cell development in the thymus [60]. During myelopoiesis, expression levels of miR-125b and miR-10a decrease with differentiation, and the corresponding increased expression levels of their target proteins may explain how key physiological differences develop between immune cells [59]. For example, miR-125b has many targets, including several transcription factors involved in B cell and T cell differentiation [59]. On the other hand, miR-10a targets transcription factors required for monocytopoeisis and megakaryocyte differentiation [59]. In mice, miR-142 has a role in maintaining cell levels of type 1 Innate Lymphoid Cells (ILCs), NK cell survival, and response to cytokines [61]. This may contribute to the greater susceptibility of miR-142-deficient mice to Murine Cytomegalovirus (MCMV) infection compared to wild-type counterparts [61]. An overview of the roles of miRNA in general immunity is provided in Figure 1. The roles of miRNA in immune cell responses to viral pathogens will be explored in-depth in the following sections. 

miRNAs add a new layer to the complexity of the immune system. The immune system is constantly in a dynamic yet delicate balance. miRNAs may provide insight into diseases of the immune system and possible treatments. This includes autoimmune diseases such as multiple sclerosis (MS) and Rheumatoid Arthritis (RA) [62], as well as allergic reactions [63] and cancers such as leukemia [57]. Population differences in miRNA expression may also inform differences in immunity [64]. With roles in both development and function, miRNAs help moderate immune responses and maintain immune cell homeostasis. 

## 5. Pro-Viral miRNA Regulation of Viral Infection

The host response to a viral pathogen is triggered at the point of virus binding and interaction with the cell. The viral entry process involves intricate interactions with host cell-specific receptor molecules. Interactions with these appropriate receptors trigger entry of the viral pathogen. For example, SARS-CoV-2 (RNA virus) and Kaposi’s sarcoma-associated herpesvirus (KSHV; DNA virus) interact with ACE2 and integrins expressed on the host cell surface, respectively [65,66]. Such interactions allow SARS and KSHV to be internalized, or in other words, establish infection. This can activate cell signaling pathways to induce cellular miRNA production and create an antiviral or pro-viral environment [67]. 

Viruses may also encode their own miRNA, known as v-miRs. The site of viral genome replication influences the method of v-miR biogenesis. Some viral genomes, such as SARS, end up in cytoplasm, while others, like KSHV, end up in the nucleus for replication. Viruses that replicate in the cytoplasm cannot access host miRNA biogenesis machinery in the nucleus [40]. However, cytoplasmic translocation of Drosha can occur during viral infection, allowing for the processing of miRNA encoded in the viral genome [68]. Retroviruses such as Human Immunodeficiency Virus (HIV) integrate their genomes into host chromosomes via reverse transcription [68]. Since their replication occurs in the nucleus, miRNA production can occur following canonical host pathways [68]. Similarly, DNA viruses that replicate in the nucleus have access to host miRNA biogenesis machinery for v-miR production [40].

Viral pathogens use miRNAs to their advantage in several ways. v-miRs can target host (cellular) or viral mRNA directly, binding to the 3′ or 5′ UTR (Figure 2) [13]. When canonically binding to the 3′UTR, they typically destabilize the transcript and trigger the degradation pathway of cellular mRNAs [69]. v-miRs have also been shown to bind to the 5′UTR, preventing translation without altering mRNA levels in the cell [70,71]. Alternatively, v-miRs binding to the 5′UTR can increase mRNA stability, preventing degradation and upregulating gene expression [14,72]. Likewise, binding to the 5′ non-translated region (NTR) can increase RNA viral genome stability [67]. v-miRs can target unique binding sites or act as functional mimics of host miRNAs by binding to the same site as corresponding endogenous miRNAs [73]. 

Viruses take advantage of pre-existing machinery within infected cells to promote viral processes. Some viruses, such as Hepatitis C Virus (HCV), do not encode viral miRNA but can regulate endogenous miRNA levels [74,75,76]. Viruses may induce the synthesis of host miRNAs when they are recognized upon entry [77]. They encode proteins that degrade a host miRNA or upregulate it [78,79,80]. Under certain circumstances, viruses exploit differential miRNA profiles between host cells to have tissue-specific effects depending on local expression levels [14,43]. This is especially important for viruses that target specific tissues, like hepatitis viruses in the liver, as well as viruses with latency phases in specific tissues, like herpesviruses in neurons [14,81,82]. Viruses can also utilize host secretory pathways and the exosomal release of miRNAs for uptake by other cells [71,81,83,84]. 

Viruses further manipulate cellular functions through v-miRs or modulation of host miRNA levels by regulating the transcription of specific genes. This includes targeting a transcription factor [85], enhancer [13,43], repressor [69], chromatin modifier [86], or another regulator within the cell [87]. In some cases, miRNAs may regulate signaling pathways within the cell to increase or decrease transcription of stimulated genes [71,88,89]. miRNAs can also be used by the virus to regulate viral gene expression and transcription of the viral genome [10,90]. v-miRs or host miRs may be utilized to promote viral replication but may alternatively downregulate it to maintain chronicity or latency [71,82,86]. The use of miRNA gives viruses the ability to control gene expression and manipulate host cell activities to their benefit. miRNAs have been implicated in many processes during infection, including viral entry, viral replication, latency and reactivation, immune evasion, immune suppression or overactivation, inflammation, host cell survival, tumorigenesis, and more [10,13,14,43,69,70,71,77,80,81,82,85,86,87,88,89,90,91,92,93,94,95,96,97,98]. See Table 1 and Table 2 for recent research developments in miRNA regulation of viral infection. 

It is important to note that miRNAs predicted to be encoded by RNA viruses have been somewhat controversial [99]. There is concern that the excision of miRNA or miRNA-like fragments could result in cleavage of the viral genome and hinder viral replication [99]. It has been suggested that the HIV encodes a transactivating response (TAR) motif that includes miRNA precursors that can be produced without cleavage of the viral genome [100,101]. However, the exact mechanism of this non-canonical pathway of miRNA production is still under investigation [100,101]. That said, v-miR-TAR has been detected in the sera of HIV-infected patients along with HIV-encoded v-miR-88 and v-miR-199 [102]. Dengue virus, Ebola virus, H5N1 Influenza virus, and SARS-CoV-2 have also been reported to encode functional miRNA-like small RNAs [13,99]. 

The discovery of virally encoded miRNA and the ability of viruses to manipulate host miRNA has the potential to elucidate new viral functions and host-virus interactions. However, some of the viral miRNA characterized in publications and their putative targets are only computer-predicted [13,98]. The production of these miRNAs and their interactions with the targets need to be verified through in vitro and in vivo studies. In addition, the outcome of these interactions should be confirmed at the cellar and organismal level. For example, miR-mRNA interactions predicted to decrease gene expression by degradation of mRNA should reflect this in both attenuated mRNA transcript levels and reduced protein output [70]. Furthermore, researchers may consider expanding on the local or systemic effects of viral miRNA packaged into exosomes and virus-induced changes in exosomal host miRNA levels [10,83,103]. They should also critically evaluate if changes in host miRNA levels are due to viral manipulation or if they are a potential defensive response to infection. 

**Table 2 biology-12-01334-t002:** Pro-viral miRNAs involved in infection by DNA viruses EBV, HSV-1, HCMV, and HBV. The hsa-miRs are encoded by the human host and may be manipulated by viruses. The miRs without hsa notation are virally encoded.

Virus Name	miRNA	Target(s)	Viral Effect	Viral Significance	Ref.
Epstein–Barr Virus (EBV)	miR-BART3, miR-BART19	RIG-1	Downregulate PRR and Type I IFN production in B cells	Innate immune evasion	[89]
	miR-BART1; miR-BART3	IRF-9; JAK1	Downregulate JAK/STAT pathway response to Type I IFN and ISGs	Innate immune evasion	[89]
	miR-BART11; miR-BART17-3p	FOXP1; PBRM1	Downregulate repressors of PD-L1 transcription to prevent T cell cytotoxic activity against infected cells	Immune evasion	[69]
	miR-BHRF1-1	p53 gene	Downregulate p53 to decrease cell cycle arrest, prevent apoptosis, and induce proliferation	Cell survival, proliferation, tumorigenesis	[91]
Herpes Simplex Virus (HSV-1)	hsa-miR-138	HSV-1 ICP0, Oct-1, Foxc1	High expression of miR-138 in neurons allows cell-specific repression of viral gene expression, transcription, and replication	Latency	[82]
	hsa-miR-24	STING	Induces the production of miR-24 to downregulate the STING pathway and decrease IFN production	Immune evasion	[77]
Human Cytomegalovirus (HCMV)	miR-US33as-5p	IFNAR1	Downregulates IFN activation of the Jak/STAT pathway and the transcription of ISGs	Immune evasion	[88]
	miR-US5-2; miR-UL22A	NAB1; SMAD3	Upregulate TGF-β production to decrease CD34+ HPC proliferation and myelopoiesis; downregulate TGF-β-stimulated genes	Myelosuppression; latency and reactivation	[92]
	miR-UL148D	ERN1	Downregulates the JNK signaling pathway and ER stress-induced apoptosis	Host cell survival	[70]
	miR-UL59, UL70-3p, US4-5p, US5-1, US22-5p, US25-2-5p, US29-5p, US33-5p	ERAP1	Viral miRNAs preferentially bind different genetic variants of ERAP1 to downregulate MHC class I antigen processing	Immune evasion	[93]
Hepatitis B Virus (HBV)	hsa-miRNA-548ah	HDAC4	Promotes miRNA-548ah expression, downregulating HDAC4 to reduce histone interactions with viral cccDNA	Viral replication; viral transcription	[86]
	HBV-miR-3	SOCS5	Upregulates the JAK/STAT pathway and ISGs in hepatocytes; exosomal HBV-miR-3 triggers macrophage polarization to M1 and promotes EGFR to increase IL-6 secretion.	Innate immune activation; maintenance of chronic infection	[71]
	hsa-miR-192-3p	ZNF143	Upregulates miR-192-3p in hepatocytes to downregulate ZNF143/Akt/mTOR signaling, enhancing viral replication	Viral transcription; viral replication	[43]

## 6. miRNA Regulation of the Antiviral Host Response

Viral pathogens manipulate host miRNA for a reason: they are master regulators of immunity. Host miRNA may target viral gene expression directly or indirectly to prevent viral replication [72,104]. Binding to the 3′ NTR of an RNA viral genome can inhibit translation and, thus, replication in some cases [67]. They are involved in the intracellular antiviral response upon infection, including the regulation of pathways for interferon production or transcription of interferon-stimulated genes (ISGs) [105]. miRNAs play a role in immune cell activation, effector functions, and memory development, as well as antigen presentation of infected cells [106,107,108,109]. They may mediate the susceptibility of host cells to infection through the regulation of endocytic or secretory pathways [106,110,111]. In addition, host cells have defenses against viral miRNAs [112]. Table 3 provides a summary of antiviral miRNAs, their targets, and their role in viral infection. 

### 6.1. miRNA in the Intracellular Antiviral Response

Cells have many mechanisms in place to guard against viral pathogens. When infection occurs, cells display different miRNA profiles compared to healthy or resistant cells [113,114]. While some miRNAs may be manipulated by the virus for pro-viral functions, others have antiviral effects [43,72,115]. 

Host miRNAs may target viral genes directly to downregulate expression. For instance, hsa-miR-3145 has been shown to silence the viral *PB1* gene in H5N1, H1N1, and H3N2 subtypes of Influenza A Virus (IAV) [72]. The miRNA-mediated downregulation of the PB1 protein prevents effective viral transcription and replication in A549 cells [72]. Type I IFN can upregulate hsa-miR-1307 during H1N1 infection of A549 cells to inhibit the expression of its target, the viral *NS1* gene [72]. NS1 protein has been implicated in creating a favorable environment for viral replication through G0/G1 cell cycle arrest of infected cells [72]. By downregulating NS1, miR-1307-3p is reported to be an effective inhibitor of IAV replication [72]. Similarly, hsa-miR-150-5p targets the *nsp10*-coding strand in the SARS-CoV-2 genome [10]. The *nsp10* gene product is important in facilitating viral replication and immune evasion [10]. As seen in Table 1, hsa-miR-150-5p downregulation in moderate–severe patients during SARS-CoV-2 infection may have pro-viral consequences [10]. 

Host miRNA may impair viral infection by interfering with viral entry. For example, during KSHV infection of human endothelial cells (HMVEC-d), hsa-miR-36 is upregulated shortly after infection and targets IFITM1, an interferon-induced transmembrane protein upregulated by KSHV that helps facilitate viral entry [25]. In this case, cells directly combat manipulation by the virus. During SARS-CoV-2 infection, resistant cell lines have higher expression of several miRNAs that target host cell receptor proteins involved in viral entry when compared to susceptible cell lines [113]. This includes ACE2, predicted to be targeted by miR-9-5p and miR-218-5p, as well as TMPRSS2 targeted by let-7d-5p, miR-494-3p, miR-382-3p, let-7e-5p, miR-181c-5p, and miR-452-5p [113]. During SARS-CoV-2 infection, hsa-miR-1827 and hsa-miR-1277-5p are predicted to target proteins involved in viral entry and antigen presentation of the viral spike protein in host cells [106]. A miRNA implicated in HCV viral entry is miR-182, which targets a tight junction protein CLDN1 that aids in the internalization of the virus [116]. mir-182 expression in infected Huh7 cells decreased viral load compared to mir-155, a known inhibitor of CLDN1 that contrastingly increased viral load [116]. 

Host cells also have methods to contend with virally encoded miRNA. Recent evidence suggests that some host circular RNAs act as sponges or decoys to prevent the binding of viral miRNAs to their targets [112,117]. For instance, hsa_circ_0001400 is induced during KSHV infection of HUVEC cells [112]. This circ-RNA is predicted to have a binding site for KSHV-encoded miR-K12-10b and was shown to decrease the expression of viral *RTA* and *LANA* genes [112]. However, this effect may be due to interactions between the circ-RNA and human mRNA encoding transcription factors involved in chromatin modification instead [112]. Host cells can alternatively regulate their own miRNA to further antiviral cellular functions. During IAV infection of A549 cells, the intronic circ-RNA AIVR is upregulated and functions as a miRNA sponge in the cytoplasm that sequesters mir-330-3p [117]. mir-330-3p downregulates CREBBP, a protein involved in enhancing IFN-β production [117]. The circ-RNA AIVR downregulates mir-330-3p activity to increase the expression of this antiviral factor [117]. 

miRNAs can be released via exocytosis and exert antiviral effects in other cells. During HIV infection, intestinal epithelial cell TLR3 activation can induce the production of miRNAs that can restrict HIV [111]. This includes miR-17 and miR-20, which downregulate the expression of p300/CBP-associated factor (PCAF), an HIV protein cofactor [111]. In addition, miR-28 targets the HIV transcript, and miR-29 family members interfere with HIV replication [111]. The uptake of these exosomal miRNAs, along with other antiviral elements were able to induce the production of HIV restriction factors in macrophages [111]. 

### 6.2. miRNA in Cellular Signaling Pathways during Viral Infection

Cellular signaling pathways involved in the antiviral response can also be regulated by host miRNA. The Wnt, IFN, MAPK, and NF-κB pathways have all been implicated in antiviral responses [118]. The Wnt pathway is associated with cell survival and proliferation and is activated during Rotavirus infection (RV) [119]. hsa-miR-192 and hsa-miR-215 target the frizzled receptors, while hsa-miR-181a directly targets β-catenin (CTNNB1) in this pathway [119]. These host miRNAs are downregulated during RV infection to promote Wnt/β-catenin signaling and survival of infected cells [119]. However, combined overexpression of miR-192 and miR-215 in RV-infected Caco2 cells inhibited RV replication [119]. The Wnt pathway is also targeted and downregulated by the miR-34 family, which has been shown to repress flavivirus replication and enhance the interferon response in infected cells [115]. 

The interferon pathway is a key player in intercellular signaling and production of antiviral effector proteins. The miR-183 cluster, consisting of miR-96, miR-182, and miR-183, promotes IFN signaling and production, and was shown to decrease vesicular stomatitis virus production in infected HepG2 cells [105]. Type I and III IFNs typically activate the JAK/STAT pathway with downstream activation of STAT1, a transcription factor regulating ISGs, but miR-183 can upregulate STAT1 mRNA without immune stimulation [105]. miR-183 also downregulates PP2A and TRIM27, two repressors of IRF3 activation, to increase Type I and III IFN production [105]. Influenza A virus has been shown to downregulate the miR-30 family due to its antiviral effects [120]. miR-30 family members downregulate SOCS1 and SOCS3, inhibitors of the IFN/JAK/STAT pathway [120]. 

The p38 MAPK pathway can be activated by many stimuli, including signaling via cytokines or viral recognition [104]. This pathway is upregulated in IAV infection, but suppression of p38 MAPK or the downstream MK2 protein can inhibit viral replication of IAV and Respiratory Syncytial Virus (RSV) [104]. miR-124a, miR-744, and miR-24 have broad antiviral effects against IAV and RSV, including downregulation of MK2 and decreased activation of p38 MAPK [104]. During coxsackievirus (CVB3) infection of HeLa cells, miR-21 was shown to be upregulated, resulting in downregulation of its target MAP2K3 and suppression of the p38 MAPK signaling pathway [110]. This inhibited viral progeny release and decreased apoptosis [110].

The NF-κB pathway has many roles in innate and adaptive immunity [118]. During Human Cytomegalovirus (HCMV) infection of neural precursor cells (NPCs), mir-221 is upregulated [121]. mir-221 directly targets and downregulates SOCS1, an inhibitor of cytokine signaling [121]. This promotes the phosphorylation and activation of NF-κB as well as the Type I IFN signaling pathway to increase inflammation and decrease HCMV replication [121]. 

### 6.3. miRNA in Immune Cell Response to Viral Pathogens

The signaling molecules released by infected cells can be regulated by miRNA and may recruit immune cells. miRNA may, in turn, regulate immune cell effector functions to fight viral infection. In pNK cells, miR-362-5p is upregulated and targets CYLD, an inhibitor of NF-κB signaling [122]. miR-362-5p upregulation of the NF-κB pathway resulted in enhanced effector functions of NK cells [122]. In MCMV infection of mice, cytokines can upregulate miR-155 in NK cells and are required for effective NK cell expansion [107]. miR-155 targeted Noxa and SOCS1 in these infected NK cells [107]. In CD8+ T cells, miR-155 is expressed highly in effector cells and is upregulated to a lesser extent in memory cells when compared to naïve counterparts [108]. Like NK cells, miR-155 targeted SOCS1 and was needed for robust expansion and cytokine signaling of CD8+ T cells during Lymphocytic Choriomeningitis (LCMV) infection of mice [108]. During Vesicular Stomatitis Virus (VSV) infection of mice, miR-155 was needed for activation and proliferation of CD4+ T helper cells as well as IL-2 and IFN-γ production [109]. miR-155 activity in CD4+ cells was also needed for activation of B cells for antibody production [109]. 

During HIV infection, IL-1β-induced miR-103/107 expression in macrophages downregulates CCR5 and prevents viral entry in these cells [114]. miR-103 is upregulated in CD4+ T cells in elite controllers of HIV, which may help to explain how these individuals maintain undetectable viral load during infection [114]. Other miRNAs have been implicated in effector functions of immune cells, but have not yet been characterized in the context of viral infection [60,123,124,125,126,127,128]. 

**Table 3 biology-12-01334-t003:** Summary of antiviral miRNAs involved in infection by viruses IAV, SARS-CoV-2, HIV-1, HCV, HCMV, KSHV, RSV, VSV, and CVB3. The hsa-miRs are encoded by the human host. Targets are cellular protein transcripts unless otherwise stated.

Virus Name	miRNA	Target(s)	Antiviral Effect	Viral Significance	Ref.
Influenza A Virus (IAV)	hsa-miR-3145	Viral *PB1* gene	Downregulates viral *PB1* protein expression	Inhibits viral replication	[72]
	hsa-miR-1307	Viral *NS1* gene	Prevents the induction of cell cycle arrest	Prevents a favorable environment for the virus	[72]
	hsa-miR-24, hsa-miR-124a; hsa-miR-744	MAPK14; Myc	Suppress downstream p38 MAPK expression and activation	Inhibit viral replication	[104]
Severe Acute Respiratory Syndrome-related Coronavirus (SARS-CoV-2)	hsa-miR-150-5p	Viral *nsp10* gene	Downregulates the activation of downstream elements nsp14 and nsp16	Decreases translation efficiency, immune evasion, and viral replication	[10]
	hsa-miR-9-5p and hsa-miR-218-5p	ACE2	Downregulate the host cell receptor for the virus	Prevent viral entry	[113]
	hsa-let-7d-5p, hsa-miR-494-3p, hsa-miR-382-3p, hsa-let-7e-5p, hsa-miR-181c-5p, and hsa-miR-452-5p	TMPRSS2	Downregulate the host cell receptor for the virus	Prevent viral entry	[113]
	hsa-miR-1827	CTSV	Downregulates the host protein that regulates virus entry	Prevents viral entry	[106]
	hsa-miR-1277-5p	CANX	Downregulates the host protein that stabilizes S protein for folding	Antigen presentation	[106]
Human Immunodeficiency Virus 1 (HIV-1)	hsa-miR-17, hsa-miR-20	PCAF	Downregulate cellular cofactor of the HIV Tat protein	Prevent viral gene expression	[111]
	hsa-miR-28, hsa-miR-29a	HIV mRNA	Downregulate viral protein production	Prevent viral replication	[111]
Hepatitis C Virus (HCV)	hsa-miR-182	CLDN1	Downregulates the host protein involved in the internalization of the virus	Prevents viral entry	[116]
Human Cytomegalovirus (HCMV)	hsa-mir-221	SOCS1	Downregulates the inhibitor of NF-κB phosphorylation and activation	Promotes cytokine signaling	[121]
Kaposi’s sarcoma-associated herpesvirus (KSHV)	hsa-miR-36	IFITM1	Downregulates cellular transmembrane protein	Prevents viral entry	[25]
Respiratory Syncytial Virus (RSV)	hsa-miR-24, hsa-miR-124a; hsa-miR-744	MAPK14; Myc	Suppress downstream p38 MAPK expression and activation	Inhibit viral replication	[104]
Vesicular Stomatitis Virus (VSV)	hsa-miR-183 cluster	PP2A, TRIM27; STAT1	Downregulate negative regulators of IRF3 phosphorylation; upregulate STAT1	Promotes interferon production	[105]
Coxsackievirus (CVB3)	hsa-miR-21	MAP2K3	Suppresses the P38 MAPK signaling pathway	Inhibits viral release	[110]

### 6.4. Considerations

Antiviral responses are not limited to immune cells. Most cells have internal defenses such as those described above that assist in the immune response. Understanding the role of miRNA in these processes and signaling pathways is complicated by the cell-specific expression and effects of miRNA. Variations between cell types can make it difficult to determine the practical applicability of in vitro research to the human body. In addition, some miRNAs can be released via exocytosis, but not all. miRNAs are not randomly incorporated into exosomes, though the sorting process is still under investigation [129,130,131]. Determining which miRNAs can be released via exocytosis is crucial for ascertaining potential effects. Furthermore, the same miRNA may exert antiviral effects against one virus but serve as pro-viral for another. For example, the IFITM1 protein can enhance the viral replication of KSHV, Epstein–Barr Virus (EBV), and Herpes Simplex Virus 2 (HSV-2) but is critical in preventing the viral entry of many RNA viruses [25].

As previously discussed, hsa-miR-36 targets IFITM1 and was demonstrated to be upregulated during KSHV infection and have antiviral effects against KSHV, EBV, and HSV-2 [25]. Yet, there are no publications available on PubMed to date indicating modified expression of miR-36 during IAV, HIV, HCV, West Nile Virus, or Dengue Virus infection. miRNA regulation of viral processes is well studied, but knowledge of the mechanisms by which cells regulate antiviral miRNA in response to specific viruses is limited. The antiviral functions of miRNAs during infection are depicted in Figure 3.

## 7. miRNA Diagnostics and Therapeutics Targeting Viral Pathogens

### 7.1. miRNA Diagnostics

Many studies have investigated the modulation of miRNA profiles during viral infections and have indicated possible biomarkers for diagnostics [132,133,134,135]. The principle of using miRNAs as diagnostics is based on the circulating levels of miRNAs in biological fluids. Current studies suggest that the expression of circulating miRNA can indicate the severity of viral infections such as SARS-CoV-2 [10,136]. In addition, miRNA profiles have the potential to predict health consequences related to viral infection. For example, levels of circulatory EBV miRNAs may predict nasopharyngeal carcinoma and a plasma miRNA panel could be utilized to screen for Hepatitis B Virus (HBV)-related hepatocellular carcinoma [137,138]. miRNAs may also be used to track disease progression and predict treatment outcomes using agents that do not target miRNA directly [139,140,141,142]. 

### 7.2. miRNA-Based Therapeutics

Because of its important role in the regulation of viral infection, miRNA has remarkable potential for antiviral therapeutics. Therapies could function by mimicking antiviral miRNA or antagonizing pro-viral miRNA [19,25,143,144]. The HCV drug miravirsen was the first antimiR-based agent administered to patients and is currently undergoing phase 2 clinical trials [19,145]. It is an antisense oligonucleotide (ASO) that targets miR-122 and has been shown to reduce HCV RNA levels in a dose-dependent and prolonged manner in a phase 2 clinical trial [145]. There was also no evidence of dose-limiting toxicity, and no patients in the trial discontinued treatment due to adverse events [145]. A different study indicates that miravirsen can effectively be used alone against HCV in vitro, including variants resistant to direct-acting antivirals, and has additive antiviral effects when used in combination with current NS3, NS5B, and NS5A inhibitors [146]. RG-101 was another antimiR targeting miR-122 shown to decrease HCV RNA levels [147,148]. However, this clinical trial was stopped due to elevated bilirubin levels in the blood [147,148]. 

There are current studies investigating the antiviral potential of other miRNAs related to HCV infection, as well as HSV, SARS-CoV-2, IAV, HIV, and other viruses that have not yet reached clinical trials [10,25,94,143,149,150,151,152,153,154,155]. Future antiviral miRNA-based therapies may include miRNA mimics, which have been used successfully in clinical trials against cancer but have not reached clinical trials in antiviral therapy [148]. Furthermore, artificial miRNA sponges and miRNA masking ASOs have been proposed as possible methods to sequester miRNA or mask the binding site of miRNA on its target, respectively [148].

In addition to therapies based on human miRNA, there are some studies investigating the use of plant miRNA for antiviral therapeutics. Treatment with honeysuckle extract has been shown to increase miRNA let-7a levels during Enterovirus infection leading to a decrease viral replication in vitro and in mice [156]. The honeysuckle-encoded atypical microRNA2911 has been studied in the context of IAV and SARS-CoV-2 infection and has predicted binding sites in both viral genomes [157,158,159]. It was shown to decrease viral replication against IAV H1N1, H5N1, and H7N9 in mice [158]. SARS-CoV-2 patients taking routine antiviral therapy given a honeysuckle decoction daily had absorbed miR2911 detected in serum, decreased viral replication, and were more likely to test negative seven days after diagnosis compared to infected individuals given routine antiviral therapy and a Traditional Chinese Medicine (TCM) mixture without miR2911 [157]. However, the authors do not disclose the ingredients of the TCM mixture, nor do they include a patient group given antiviral therapy without honeysuckle decoction or TCM [157]. A later study found that patients with a SIDT1 polymorphism had decreased absorption of miR-2911, and the honeysuckle decoction failed to inhibit SARS-CoV-2 replication [159]. This does not contradict the previous findings concerning the antiviral effects of miR-2911. Instead, it underscores an important consideration when evaluating oral treatments and absorption of dietary miRNAs.

Chimeric Antigen Receptor (CAR)-T cell therapy has shown promise in treating cancers and has recently been studied as a potential therapy for autoimmune diseases and viral infections [160]. This form of immunotherapy uses human T cells engineered to produce CARs with the ability to bind to specific proteins and target cancerous or diseased cells [161]. The application of CAR-T cells in infectious diseases is currently under investigation, with promising results in treating HBV, HCV, HCMV, HIV, and SARS-CoV-2 [160]. In addition, the chimeric autoantibody receptor (CAAR) T cells have been used to treat autoimmune diseases by targeting B cells with specificity for autoantigen receptors [160]. Since miRNAs regulate cellular activities without complete gene knock-out, they can be used to fine-tune the design of CAR-T cells and help mitigate their limitations. For example, miR-155 upregulation in CAR-T cells has been shown to promote T cell function, survival, and infiltration [161,162,163]. Similarly, the upregulation of miR-H18 in CAR-T cells has been shown to enhance cytotoxic activity [164]; miR-17-19 has been used to promote the persistence of effector T cells [161]; and the overexpression of miR-27a-5p may enhance the infiltration of tissues [163]. Furthermore, the overexpression of miR-143 promotes memory T cell formation for enhanced specificity and reduced toxicity in CAR-T therapy [161]. This research reveals miRNA as a crucial tool for improving current therapeutics.

### 7.3. Challenges and Future Considerations

Several miRNAs have been implicated in the regulation of viral infection. However, in many cases, their exact mechanisms have yet to be elucidated [96,98]. Computational approaches may be used to predict miRNA targets but can have high false-positive rates [165]. Novel algorithms continue to be developed in efforts to improve these predictions [165,166,167,168,169,170,171]. Predicting miRNA binding sites can cut down on research costs by narrowing potential targets, but they need to be experimentally validated using in vitro and in vivo models [165]. 

Different strains of viruses can cause similar changes in miRNA levels, with some variations partly due to minor changes to their antigenicity and replication. For example, the Mayinga, Makona, and Reston strains of the Ebola virus share 82% early-phase miRNAs and 90% late-phase miRNAs, and only seemed to selectively modulate one out of five miRNAs [172]. Another study found that only 5 out of 25 differentially expressed miRNAs were similarly modulated across four Lyssavirus strains [173]. A viral protein from Epstein–Barr Virus (EBV) has also been shown to distinctly modulate the expression of 3 miRNAs between M81 and B95.8 strains [174]. An in silico study on SARS-CoV-2 found redundancy in predicted pre-miRNAs encoded in the viral genomes of different strains, with six unique pre-miRNAs between the Reference (Wuhan), Delta, and Omicron strains [175]. These subtle variations could be used to explain differences in the pathology of different strains or to identify individual strains for diagnostic purposes. Alternatively, targeting miRNAs shared across multiple strains could be beneficial when developing therapeutics with broad applications. 

miRNA-based diagnostics rely on characteristic miRNA profiles for given diseases [19]. However, miRNA profiles naturally show variation between individuals, and not every virus causes a characteristic upregulation of a tissue-specific miRNA such as miR-122 in HCV [14,64]. Diagnostic tools scanning for multiple dysregulated miRNAs are currently being developed for non-viral diseases, but similar panels could be used for viral infections [19]. Testing for virally encoded miRNAs may be even more effective in diagnosing viral infections than running an endogenous miRNA panel in some cases [137]. However, not all viruses are known to encode miRNAs, and some predicted v-miRs are still under investigation [13,68]. 

miRNA-based therapeutics have the capacity to combat viral infections. However, they face three main challenges: delivery, specificity, and tolerance [148]. For miRNA therapeutics to exert any effect, they must remain stable and be delivered efficiently to the desired cell type [148]. Extracellular miRNAs have a relatively short half-life in mouse serum ranging from ~1.5 h to more than 13 h depending on the sequence [176]. Because miRNA is naturally occurring in cells, there are degradation mechanisms already in place [148]. This can be beneficial when considering toxicity, but chemical modifications may be needed to extend the half-life of miRNA-based therapeutics [148,176]. RNA is negatively charged and does not passively diffuse across cell membranes, so delivery methods must facilitate cell entry [148]. Exosomes, liposomes, antibody conjugates, viral vectors, and various nanoparticles have been studied as delivery vehicles for RNA-based therapeutics [177]. Exosomes are particularly enticing as delivery vehicles of miRNA because they are natural, can protect the contents from enzymatic degradation, fuse directly with the cell membrane, are unlikely to evoke an immune response, can contain multiple compounds for combined therapeutic effects, and could be modified to have enhanced uptake by targeted cell types [148].

Some miRNAs can bind to multiple targets with similar seed sequences [148]. Therefore, specificity is essential in preventing off-target effects. Targeting genes unique to the virus using miRNA-like sequences may allow for direct and specific inhibition of viral gene expression. Furthermore, researchers may consider adding an element to control for tissue specificity. For example, the injection of artificial pri-miRNA transcripts based on human pri-miR-31 with liver-specific promotors targeted the X gene sequence in HBV circular DNA and was shown to decrease viral replication in mice [178]. Virus-specific considerations also need to be accounted for. For example, treatment of HBV infection is made difficult by the persistence of viral covalently closed circular DNA (cccDNA), which was combated by targeting the open reading frame (ORF) encoding the HBx protein that maintains the cccDNA [178,179]. On the other hand, some miRNA-based therapies may be enhanced by less specificity, such as a miRNA sponge that could sequester multiple types of pro-viral miRNA [117,180]. 

For miRNA-based therapies to be safe and effective, they must be tolerated well by the body [148]. This includes using safe delivery vehicles that do not mount an immune response as well as limiting off-target toxicity [148]. In addition, the miRNA therapeutic itself should not have cytotoxic effects or induce immune-related adverse events [148]. Immunogenicity can be reduced by avoiding GU-rich sequences which are recognized by TLRs, using dsRNA rather than ssRNA to reduce the likelihood of an immune response, using sequences less than 21 bp for ssRNA to minimize activation of TLRs, and adding modifications to prevent interactions with proteins such as TLRs or degrading enzymes [148]. Any chemical modifications added to enhance miRNA-based therapeutics should emulate natural RNA to prevent increased risk of off-target effects [177]. 

In miRNA-based therapeutics, researchers must be wary of varying effects in different cell types, particularly when manipulating levels of miRNA with multiple targets [148]. This is highly relevant when considering possible side effects. For example, miravirsen targets miR-122 in HCV patients and lowers HCV replication but can also lower cholesterol levels [145]. However, less specificity may not be a disadvantage in some cases and may lead to miRNA-based therapies with broad antiviral effects that have the potential to treat multiple viruses [25,104]. Further advances in miRNA therapeutics should aim to improve delivery, stability, tolerability, and specificity for appropriately targeted and prolonged effects that minimize toxicity and do not evoke an immunogenic response [148,155]. Nutritional uptake of miRNAs provides another avenue for immune modulation and may be of interest to researchers investigating therapeutics or possible active ingredients in homeopathic remedies [158,181]. In addition, diagnostic and therapeutic research should consider population differences in miRNA expression related to immunity and the immune response to viral infections [64]. This may provide insight into how to make therapeutics based on resistant populations or avoid sampling biases in diagnostic devices [182,183]. 

## 8. The miRNA Market Landscape

With several potential applications of miRNAs, the global miRNA market size has been increasing at a rapid pace. In 2022, it was valued at around 1230.3 million USD and is expected to grow at a CAGR (compound annual growth rate) of 19.9% from 2022 to 2032 [184]. Among the various subspecialties, oncology has the largest market share at 33%, followed by infectious diseases [185]. In terms of applications, the disease diagnostics segment accounted for the largest market share of 54% in 2019. miRNAs are among the most promising biopharmaceutical candidates making their way into the commercial world for the development of future medications [186]. Examples of a few companies that hold the market share for miRNA applications are Qiagen, Thermo Fisher Scientific, PerkinElmer, Illumina, Takara Bio, Mirna Therapeutics, miRagen Therapeutics, Regulus Therapeutics, and Santaris Pharma [155].

## 9. Conclusions

This review offered an overview of miRNA involvement in immune processes related to viral infection as well as significant advancements and potential applications of miRNA in the field of viral diagnostics and therapeutics. The differential expression of miRNA profiles can act as potential biomarkers for future diagnostic purposes, to predict the severity of viral infections, or to assess adverse health effects associated with viral diseases. miRNA-based diagnostic tools are useful in detecting and measuring specific miRNAs indicative of viral infections. The novel approach of using miRNA mimics to enhance antiviral activity or antagonists to inhibit pro-viral miRNAs is still under investigation but is showing promise. Plant miRNA has also been studied for use in viral treatments but requires further clinical experiments.

Several challenges, as outlined above, need to be addressed to fully harness the potential of miRNA in medicine. Delivery methods must be optimized to ensure efficient and well-targeted delivery of miRNA to the cells of interest. Specificity is key for avoiding off-target effects, and methods for enhancing tissue specificity and targeting viral genes are still being explored. The cell-specific nature of miRNA expression adds complexity to this process as well. Additionally, miRNA-based therapeutics should be well tolerated by the body, and efforts must be made to overcome cytotoxicity and possible immune-related adverse effects. Treatment methods should consider the pros and cons of the administration route and absorption of miRNA. Researchers need to be conscious of miRNA’s ability to act as an antiviral or pro-viral element when considering therapeutic uses. Population variations in miRNA expression levels warrant further study as they may shed light on differences in immunity and treatment viability. 

miRNA technology offers an exciting new approach for future diagnostics and therapeutics against viral infections. Further investigations should attempt to decipher the potential effects of virally encoded miRNAs on the host system. Endogenous miRNAs with antiviral properties should continue to be evaluated for their therapeutic potential. The journey toward refining recently discovered techniques to address existing challenges and ensure their safety, efficacy, and clinical applicability is a long one. Continued advancements in this field show great promise for the future of miRNA-based medicine.

## Figures and Tables

**Figure 1 biology-12-01334-f001:**
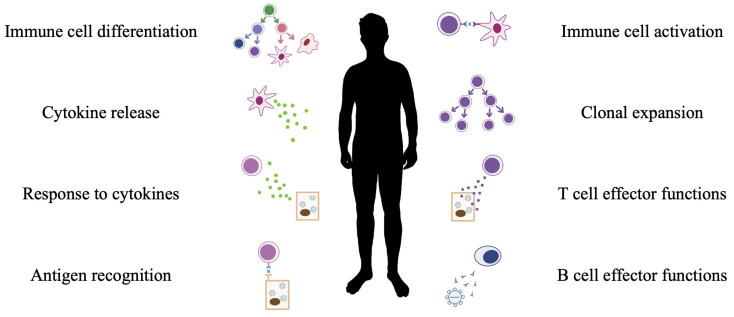
Summary of miRNA regulated processes in immunity.

**Figure 2 biology-12-01334-f002:**
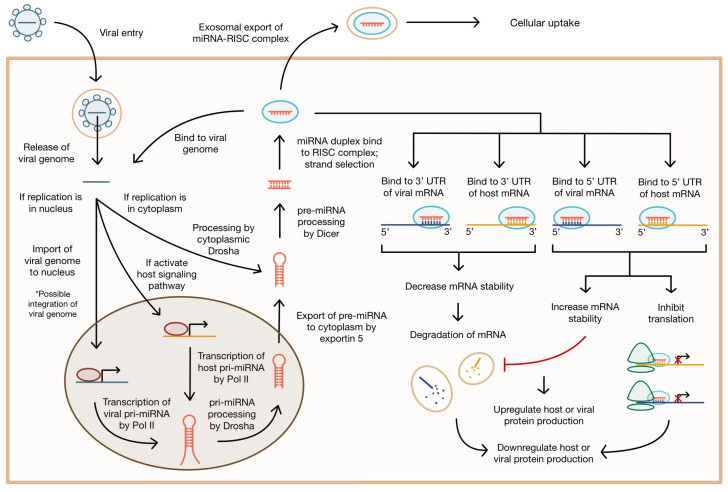
Biogenesis of miRNA during viral infection and possible effects. miRNA may be encoded by the host or virus. Viruses may upregulate or downregulate endogenous miRNA. miRNA may bind the viral genome, mRNA, or be packaged in an exosome for export to other cells. The binding site of miRNA at the 3′ or 5′ UTR of its mRNA target may alter stability of mRNA or inhibit translation.

**Figure 3 biology-12-01334-f003:**
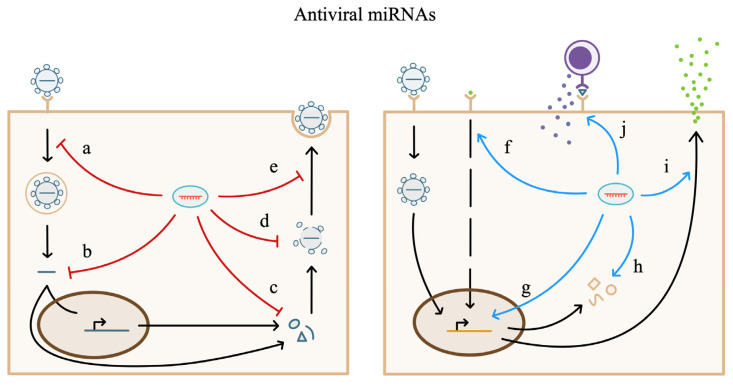
Antiviral miRNAs inhibit viral replication (**left**) including (a) viral entry, (b) transcription of the viral genome, (c) viral protein production, (d) viral assembly, and (e) viral escape. Antiviral miRNAs enhance antiviral cellular processes (**right**) including (f) cell signaling pathways (g) transcription of ISGs, (h) antiviral cellular protein production, (i) interferon release, and (j) immune cell recognition. Pro-viral miRNAs have opposing effects.

**Table 1 biology-12-01334-t001:** Pro-viral miRNAs involved in infection by RNA viruses IAV, SARS-CoV-2, HIV-1, and HCV. The hsa-miRs are encoded by the human host and may be manipulated by viruses. The miRs without hsa notation are virally encoded.

Virus Name	miRNA	Target(s)	Viral Effect	Viral Significance	Ref.
Influenza A Virus (IAV)	hsa-miR-132-3p	IRF1	Upregulates miR-132-3p to inhibit Type I IFN production and downregulates ISGs	Immune evasion, host cell survival, and viral replication	[94]
	put-hsa-miR-34	STAT3	Downregulates STAT3/IL-6-mediated antiviral response and upregulates the NF-κB pathway	Viral replication; prevention of immune and inflammatory responses	[87]
Severe Acute Respiratory Syndrome-related Coronavirus (SARS-CoV-2)	MR-147-3p	EXOC7; RAD9A; TFE3	Downregulates exocytosis; cell death and apoptosis; lipid and glucose metabolism, and TGF-β-induced transcription	Exocytosis; host cell survival; metabolism and transcription of host genes	[13]
	MR359-5p	FOXO3; GCPR1	Downregulates autophagy and dysregulates oxidative damage responses; binds 5′UTR to upregulate GPCR1 and viral propagation	Host cell survival; viral pathogenesis	[13]
	hsa-miR-148a; hsa-miR-590	USP33; IRF9	Higher exosomal loading of miR-148a and miR-590 in infected cells; downregulate USP33 and IRF9 in macrophages to upregulate NF-kB, TNFα, and IFNβ pathways	Hyperinflammation	[95]
	hsa-miR-150-5p	Viral nsp10 gene	Downregulates miR-150-5p to prevent decreased viral gene expression	Translational efficiency, viral replication, and immune evasion	[10]
Human Immunodeficiency Virus 1 (HIV-1)	hsa-miR-144	Nrf2	Upregulates miR-144 to downregulate antioxidant response and impair alveolar macrophage phagocytosis	Immune evasion	[85]
	hsa-miR-210-5p	TGIF2	Induces miR-210-5p production to downregulate TGIF2 and promote G2 cell cycle arrest	Cell cycle arrest	[80]
Hepatitis C Virus (HCV)	hsa-miR-122	TLR7	Induces host miRNA and exosomal transport to macrophages to activate TLR7, inducing the NF-κB pathway and upregulating B cell activating factor (BAFF)	Autoimmune response	[81]
	hsa-miR-122	HCV genome	Liver-specific miRNA increases the stability of the viral genome and promotes viral translation	Viral replication and gene expression	[14]

## Data Availability

No new data were created or analyzed in this study. Data sharing is not applicable to this article.

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
