# Peer review of "MicroRNAs: Small but Key Players in Viral Infections and Immune Responses to Viral Pathogens"

_biology, 2023, doi:10.3390/biology12101334_

Round 1
Reviewer 1 Report
This systematic review provides a thorough review of the role of miRNAs in immune responses, especially to viral pathogens as well as a discussion of thier potential use in diagnostics and therapeutics. It is a great review for researchers who would like to understand this rapidly evolving field.
I am not qualified enough in the field of miRNAs to comment on the thoroughness and veracity of the literature review.
Author Response
REVIEWER 1:
It is a great review for researchers who would like to understand this rapidly evolving field.
Authors Response: We thank the reviewer for the encouraging comments. We sincerely appreciate it.

Reviewer 2 Report
This review paper provides a detailed summary of microRNA's (miRNA) role in regulating viral infection, host immune responses, and miRNA-based diagnostics and therapeutics targeting viral pathogens. I have only a minor comment on this
As we know, there are different strains of the virus listed in Table 1, such as SARS-CoV-2. Is the role of miRNAs in viral infection similar for different strains? The miRNA regulation of SARS-CoV-1 pathogens can also be discussed.
Author Response
REVIEWER 2:
As we know, there are different strains of the virus listed in Table 1, such as SARS-CoV-2. Is the role of miRNAs in viral infection similar for different strains? The miRNA regulation of SARS-CoV-1 pathogens can also be discussed.
Authors Response: We appreciate the critical reviewer’s comment on the effects of plausible variations in the host responses to differences in viral strains. We have incorporated the following in section 6.3 of the manuscript (lines 483-496):
“Different strains of viruses can cause similar changes in miRNA levels with some variations partly due to minor changes to their antigenicity and replication. For example, the Mayinga, Makona, and Reston strains of Ebola virus share 82% early phase miRNAs and 90% late phase miRNAs and only seemed to selectively modulate 1 out of 5 miRNAs.[163] Another study found that only 5 out of 25 differentially expressed miRNA were similarly modulated across four Lyssavirus strains.[164] A viral protein from Epstein Barr Virus (EBV) has also been shown to distinctly modulate the expression of 3 miRNAs between M81 and B95.8 strains.[165] An in silico study on SARS-CoV-2 found redundancy in predicted pre-miRNAs encoded in the viral genomes of different strains, with 6 unique pre-miRNA between the Reference (Wuhan), Delta, and Omicron strains.[166] These subtle variations could be used to explain differences in pathology of different strains or to identify individual strains for diagnostic purposes. Alternatively, targeting miRNA shared across multiple strains could be beneficial when developing therapeutics with broad applications.”

Reviewer 3 Report
Bauer et al provide a comprehensive review of microRNA mediated regulation of viral pathogenesis. The review article is well-written and organized. I have a few comments to be included before acceptance:
1. Figure page 2: microRNA regulates gene expression and it could achieve via two modes: a. short sequence complementarity that leads to polydeadenylation or translation inhibition by recruiting several poteins of the RISC. b. perfect complementarity that leads to cleavage by AGO2 protein. This figure needs to be corrected for clarity
2. As point 1 is concerned- how V-miRs regulate gene expression (a separate table will be nice)- based on partial sequence complementarity or by cleavage. What is known about this and the mode of gene regulation?
3. Page 558-569: from where authors got this data (citation required).
4. Is there a known host microRNA that binds to viral RNA and regulates infection outcome?
Author Response
REVIEWER 3:
- Figure page 2: microRNA regulates gene expression and it could achieve via two modes: a. short sequence complementarity that leads to polydeadenylation or translation inhibition by recruiting several poteins of the RISC. b. perfect complementarity that leads to cleavage by AGO2 protein. This figure needs to be corrected for clarity
Authors Response: We thank the reviewer for this suggestion. The graphical abstract on page 2 demonstrates how miRNA influences infection following the canonical pathway (a). We focused on mRNA degradation as it is more common than translation inhibition. Perfect complementarity (b) is rare in humans, as stated in the introduction (line 83), and is more often found in plants. We have made modifications for clarity.
- As point 1 is concerned- how V-miRs regulate gene expression (a separate table will be nice)- based on partial sequence complementarity or by cleavage. What is known about this and the mode of gene regulation?
Authors Response: We found that the location of viral miRNA binding within the target mRNA was the most likely determinant of the mode of gene regulation (see first paragraph of section 4 and Figure 2 (previously Figure 1) on page 9). We have provided a footnote under the Tables that may help differentiate V-miRs from cellular miRNAs. Therefore, we did not create a separate Table. We pray that the reviewer can kindly accept such a revision.
- Page 558-569: from where authors got this data (citation required).
Authors Response: We thank the reviewer for this comment and have incorporated proper citations in the revised version of the manuscript (Page 16; lines 562-572).
- Is there a known host microRNA that binds to viral RNA and regulates infection outcome?
Authors Response: There are several known host miRNAs that bind to viral RNA, including hsa-miR-150-5p that was described recently by the studies conducted in our lab. This miRNA targets the nsp10 gene within the SARS-CoV-2 genome to lower infection of cells. Further information can be found in the second paragraph of section 5.1 (Page 9 lines 264-278).

Reviewer 4 Report
Bauer and colleagues provided a comprehensive summary of microRNAs involved in viral infections and the immune response. Their review begins with an exploration of the host response to viral pathogens, followed by an examination of microRNA regulation in the context of the host's immune response. Lastly, they delve into the potential applications of microRNAs in diagnostics and therapeutics. Despite presenting a lot of information, the organization of the content appears somewhat unstructured. Therefore, it is advisable for the authors to diligently categorize the sections, particularly when discussing how microRNA regulation is implicated in the immune response to viral infections. Furthermore, incorporating more illustrations is recommended to enhance readers' understanding of the subject matter.
1. The authors should consider simplifying the introduction of innate and adaptive immune systems in the section titled ‘2. Host response to vital pathogens’. Just describe how the immune system functions after infection by a viral pathogen, highlighting the involvement of microRNAs in this process. Additionally, incorporating an illustration would be beneficial to enhance understanding.
2. The left part of Figure 1, which explains the generation of microRNAs after viral infections, lacks sufficient description in the section titled '4. MicroRNA Regulation of Viral Infection.' This section serves as the initial connection between viral pathogen infection and microRNAs. The authors should consider reorganizing the sections for better coherence. In the 'Introduction' section, they introduce microRNAs, discussing their generation from transcription in the nucleus and their impact on mRNA stability and translation. However, this information seems to partially overlap with Figure 1 but they are placed far apart.
3. The authors have presented two tables displaying miRNAs involved in viral infections caused by DNA viruses and RNA viruses. It is essential for the authors to discuss the variations in miRNA performance when these two types of viruses infect host cells.
4. The author should provide a description of antiviral and pro-viral microRNAs before introducing them abruptly in the table. This would help readers understand the concepts and their relevance within the context of the table.
5. It is recommended that the author includes illustrations in order to enhance the comprehension of antiviral and pro-viral microRNAs mechanisms. Additionally, a summary table for antiviral microRNAs should be incorporated. To provide clear and coherent information to facilitate better reader understanding, a comprehensive reorganization of all sections is necessary to effectively categorize microRNAs into anti- and pro-viral types.
6. The authors should consider summarizing the strategies and corresponding principles used in miRNA diagnostics when discussing this topic. This would enhance the understanding of the various approaches employed in miRNA diagnostics.
7. The authors are encouraged to discuss the role of miRNA in CAR-T therapy, a crucial strategy in cancer treatment as an immunocellular therapeutic approach. This discussion would provide valuable insights into the potential impact of miRNA in enhancing the effectiveness of CAR-T therapy against cancer.
Line 20: ‘miRNA technology’ should be replaced with ‘MiRNA technology’.
Line 336, ‘is has been’ should be replaced with ‘has been’.
Line 412, ‘in resulted in’ should be replaced with ‘is resulted in’.
In Figure 1, the label "Uptake by other cells" should be revised to "Uptaken by other cells" to accurately describe the process depicted in the figure.
To ensure consistency, it is better to use the same term throughout. In this case, the authors alternated between using "miRNA" as a short form and "microRNA" in other instances.
In order to enhance clarity and facilitate readers' comprehension upon their initial encounter in the article, the authors are advised to present the complete names for abbreviations such as RV (Respiratory Virus) and IAV (Influenza A Virus).
Author Response
REVIEWER 4:
- The authors should consider simplifying the introduction of innate and adaptive immune systems in the section titled ‘2. Host response to vital pathogens’. Just describe how the immune system functions after infection by a viral pathogen, highlighting the involvement of microRNAs in this process. Additionally, incorporating an illustration would be beneficial to enhance understanding.
Authors Response: We thank the reviewer for their suggestions and have simplified this section, focusing on miRNA in immunity. We have added Figure 1 for additional understanding on page 5.
- The left part of Figure 1, which explains the generation of microRNAs after viral infections, lacks sufficient description in the section titled '4. MicroRNA Regulation of Viral Infection.' This section serves as the initial connection between viral pathogen infection and microRNAs. The authors should consider reorganizing the sections for better coherence. In the 'Introduction' section, they introduce microRNAs, discussing their generation from transcription in the nucleus and their impact on mRNA stability and translation. However, this information seems to partially overlap with Figure 1 but they are placed far apart.
Authors Response: We thank the reviewer for the suggestion to elaborate on Figure 1 (Figure 2 in the revised manuscript). We have incorporated the following to explain V-miR biogenesis (lines 159-176).
“The host response to a viral pathogen is triggered at the point of virus binding and interaction with the cell. The viral entry process involves intricate interactions with host cell specific receptor molecules. Interactions with these appropriate receptors trigger entry of the viral pathogen. For example, SARS-CoV-2 (RNA virus) and Kaposi's sarcoma-associated herpesvirus (KSHV; DNA virus) interact with ACE2 and integrins expressed on the host cell surface, respectively [54,55]. Such interactions allow SARS and KSHV to be internalized; or in other words, establish infection. This can activate cell signaling pathways to induce cellular miRNA production and create an antiviral or pro-viral environment. [56]
Viruses can also encode their own miRNA, known as V-miRs. The site of viral genome replication influences the method of V-miR biogenesis. Some viral genomes, such as SARS, end up in cytoplasm while others, like KSHV, end up in the nucleus for replication. Viruses that replicate in the cytoplasm cannot access host miRNA biogenesis machinery in the nucleus.[57] However, cytoplasmic translocation of Drosha can occur during viral infection, allowing for processing of miRNA encoded in the viral genome.[58] Retroviruses such as HIV integrate their genomes into host chromosomes via reverse transcription.[58] Since their replication occurs in the nucleus, miRNA production can occur following canonical host pathways.[58] Similarly, DNA viruses that replicate in the nucleus have access to host miRNA biogenesis machinery for V-miR production.[57]”
The paragraphs that follow explain how viruses can activate host pathways to create a more favorable environment.
We also agree with the reviewer that the placement of the figure is not ideal. Though the introduction does introduce miRNA biogenesis and function, we do not specifically address this in the context of viral infection until later sections. The subject of Figure 2 (previously Figure 1) is more of a summary for sections 4 and 5, so has been moved to the end of section 4 (page 9).
- The authors have presented two tables displaying miRNAs involved in viral infections caused by DNA viruses and RNA viruses. It is essential for the authors to discuss the variations in miRNA performance when these two types of viruses infect host cells.
Authors Response: We agree with the reviewer that the differences between DNA viruses and RNA viruses need to be addressed. We have included this information at the beginning of section 4 (lines 159-176).
- The author should provide a description of antiviral and pro-viral microRNAs before introducing them abruptly in the table. This would help readers understand the concepts and their relevance within the context of the table.
Authors Response: We appreciate the reviewer’s suggestion and have introduced antiviral and pro-viral miRNA at the end of the first paragraph in section 4 (line 166).
- It is recommended that the author includes illustrations in order to enhance the comprehension of antiviral and pro-viral microRNAs mechanisms. Additionally, a summary table for antiviral microRNAs should be incorporated. To provide clear and coherent information to facilitate better reader understanding, a comprehensive reorganization of all sections is necessary to effectively categorize microRNAs into anti- and pro-viral types.
Authors Response: We thank the reviewer for their suggestion for more illustrations and agree that they would enhance comprehension. Accordingly, significant changes have been made to the manuscript. We have added a new schematic (Figure 3) on page 13 for the antiviral and pro-viral effects miRNA can have on host cells. We have also added Table 3 (page 12) as a summary of antiviral miRNAs described in Section 5. We have organized Section 4 to cover pro-viral miRNA either V-miRs encoded by viruses or host miRNAs taken advantage of by the virus. We have renamed this section as “Pro-viral miRNA Regulation of Viral Infection” for clarity. Section 5 covers the role of miRNA in the host antiviral response including effects on the intracellular antiviral response, cell signaling pathways, and immune response. We have renamed this section “MiRNA Regulation of the Antiviral Host Response” for clarity.
- The authors should consider summarizing the strategies and corresponding principles used in miRNA diagnostics when discussing this topic. This would enhance the understanding of the various approaches employed in miRNA diagnostics.
Authors Response: We agree that the earlier version of this section was confusing. In the revised version of the manuscript, we have outlined the principle and the approaches to keep it to the point and make it simple to the readers in lines 407-409.
- The authors are encouraged to discuss the role of miRNA in CAR-T therapy, a crucial strategy in cancer treatment as an immunocellular therapeutic approach. This discussion would provide valuable insights into the potential impact of miRNA in enhancing the effectiveness of CAR-T therapy against cancer.
Authors Response: We thank the reviewer for this suggestion and agree that this will add valuable insights. We have incorporated a paragraph on the role of miRNA in improvement of CAR T therapy in section 6.2 (lines 457-474).
“Chimeric Antigen Receptor (CAR)-T cell therapy has shown promise in treating cancers and has recently been studied as a potential therapy for autoimmune diseases and viral infections.[150] This form of immunotherapy uses human T cells engineered to produce CARs with the ability to bind to specific proteins and target cancerous or diseased cells.[151] The application of CAR-T cells in infectious diseases is currently under investigation with promising results in the treatment of HBV, HCV, HCMV, HIV, and SARS-CoV-2.[150] In addition, the chimeric autoantibody receptor (CAAR) T cells have been used to treat autoimmune diseases by targeting B cells with specificity for autoantigen receptors.[150] Since miRNAs regulate cellular activities without complete gene knock-out, it can be used fine tune the design of CAR-T cells and help mitigate their limitations. For example, miR-155 upregulation in CAR-T cells has been shown to promote T cell function, survival, and infiltration.[151-153] Similarly, upregulation of miR-H18 in CAR-T cells has been shown to enhance cytotoxic activity [154]; miR-17-19 has been used to promote persistence of effector T cells [151]; and overexpression of miR-27a-5p may enhance infiltration of tissues [153]. In addition, overexpression of miR-143 may promote memory T cell formation for enhanced specificity and reduced toxicity in CAR-T therapy.[151] This research reveals miRNA as a crucial tool for improvement of current therapeutics.”
Comments on the Quality of English Language
Line 20: ‘miRNA technology’ should be replaced with ‘MiRNA technology’.
Authors Response: We have made appropriate changes in the revised version of the manuscript. We have fixed capitalization in lines 22, 28, 147, 253, 307, 350, 382, 392, 405, 416, 579, 596.
Line 336, ‘is has been’ should be replaced with ‘has been’.
Authors Response: We have made appropriate changes in the revised version of the manuscript (line 272).
Line 412, ‘in resulted in’ should be replaced with ‘is resulted in’.
Authors Response: We have made appropriate changes in the revised version of the manuscript. (line 342)
In Figure 1, the label "Uptake by other cells" should be revised to "Uptaken by other cells" to accurately describe the process depicted in the figure.
Authors Response: We thank the reviewer for the corrections and have implemented them in the revision. “uptake by other cells” has been changed to “cellular uptake” for clarity in what is now Figure 2 on page 9.
To ensure consistency, it is better to use the same term throughout. In this case, the authors alternated between using "miRNA" as a short form and "microRNA" in other instances.
Authors Response: We thank the reviewer for their suggestion. We have changed microRNA to miRNA as requested (lines 562, 567, 570 and section titles 3, 4, 5, 6, 7), with the exception of the first mention accompanied by abbreviation in parentheses in the abstract, summary, and introduction.
In order to enhance clarity and facilitate readers' comprehension upon their initial encounter in the article, the authors are advised to present the complete names for abbreviations such as RV (Respiratory Virus) and IAV (Influenza A Virus).
Authors Response: We thank the reviewer for their suggestions. We have presented complete names for abbreviations when they are first mentioned (lines 11-12, 129, 130, 141, 143, 150, 162, 221, 266, 318, 346, 360, 361, 387, 413, 488).

Reviewer 5 Report
Authors present a rather comprehensive review of miRNAs in viral infections and immune response against viruses. Initial part comprises a recapitulation of basic immunology and requires significant changes and most importantly should be shortened. There is high discrepancy in the quality of immune and miRNA parts - the latter is of much higher quality. Still, the micro RNA part is mostly chaotic and difficult to follow. Text should be re-organized, shortened and mostly re-written.
1. Lines 102-103: this sentence is misleading. Innate immunity recognises only some conserved antigens. I would suggest rephrasing this sentence so that recognition of self/non-self would be ommitted.
2. Fragment about breast-feeding should be slightly rewritten in a more concise manner with more details e.g. instead of general information please directly write that IgA can be found in human milk.
3. Human immune-competent cells develop during pregnancy. At birth every subset is present. Innate cells are fully developed, while T and B cells require further maturation.
4. NK and ILCs are considered as separate subsets. ILCs contain a fraction of NK-like cells that have however different immunophenotype.
5. " NK T-cells (NKT) are primarily involved in cytokine signaling but are from a defined T cell lineage with low specificity" - this sentence is unclear. Human T cells are usually divided into conventional (abT helpers and cytotoxic cells) and unconventional (MAIT, iNKT and gamma-delta T). What is mentioned above are iNKT (invariant NKT!). Both iNKT and gdT cells are viewed as a link between innate and adaptive immunity - they recognise some specific antigens e.g. sphingolipids for iNKT cells, phosphoantigens for Vd2 gamma delta T or B12 metabolites for MAIT cells. They are capable of rapid response with high amount of cytokines released per cell thus they are responsible for directing the immune response towards Th1/Th2/Th17 pathway.
6. T follicular helper cells have different abbreviation - Tfh
7. Lines 145-163 are somewhat misleading. It should be clearly stated that naive B and T cells that recognise antigen presented by APC get activated and undergo further maturation etc. In general the immune introduction should be significantly shortened. This is only a introductory part and should not be 5 pages long. The majority of that part repeats basic immunology for university textbooks - this is completely unnecessary and should be ommitted.
8. Lines 236-239: this is an oversimplification. It is not only about the activation/suppression balance, but also cytokine balance etc. I would suggest ommitting this sentence.
9. Lines 343: while HMVEC are normal endothelial cells from microvasculature, BJAR are Burkitt lymphoma cells. Thus results obtained for BJAB may or may not be relevant to healthy human B cells.
10. Lines 346-347: was this during infection (so in vivo) or on cell lines (so in vitro). Those are two separate things. In vitro experiments on cell lines does not replicate the complicated nature of real infection and the network of interactions between virous tissues.
11. "This includes miR-17 and miR-20, which target an HIV protein, as well as miR-28, which targets the HIV transcript, and miR-29 family members, which interfere with HIV replica" - the sole principle of micro RNA is that it targets RNA, not protein.
12. "The p38 MAPK pathway can be activated by cytokines or viral infection." - this pathway is activated downstream of numerous receptors e.g. as part of TCR signalling in T cells.
13. At least part of the mechanisms mentioned in text should be also represented as a figure.
14. Simoa is a bead based method to measure various analytes, including direct measurment of some micro RNAs.
15. Part 8. Discussion should be ommitted - it merely repeats what was said above.
16. In abstract and conclusions, authors claim that this is a SYSTEMATIC review. There is no sign however. How was it performed e.g. what search querries, databases etc were used? Which studies were included and which were excluded?
Requires significant changes
Author Response
REVIEWER 5:
- Lines 102-103: this sentence is misleading. Innate immunity recognises only some conserved antigens. I would suggest rephrasing this sentence so that recognition of self/non-self would be ommitted.
Authors Response: We thank the reviewer for their comment and have rephrased to “Innate immunity is often referred to as the first line of defense and recognizes conserved antigens including common motifs in pathogens. [36]” lines 109-110.
- Fragment about breast-feeding should be slightly rewritten in a more concise manner with more details e.g. instead of general information please directly write that IgA can be found in human milk.
Authors Response: We thank the reviewer for their comment and have rewritten the fragment in question. We have limited it to one sentence: “Human breast milk helps strengthen a newborn’s immune system and delivers ~1400 miRNAs, including many linked to immune system maturation and viral defenses.” (lines 113-115).
- Human immune-competent cells develop during pregnancy. At birth every subset is present. Innate cells are fully developed, while T and B cells require further maturation.
Authors Response: We thank the reviewer for their comment. The section about immunity has been shortened to be more concise. The section about immune cell development has been deleted.
- NK and ILCs are considered as separate subsets. ILCs contain a fraction of NK-like cells that have however different immunophenotype.
Authors Response: We thank the reviewer for their comment. The section about immunity has been shortened to be more concise. The section about ILCs has been deleted.
- " NK T-cells (NKT) are primarily involved in cytokine signaling but are from a defined T cell lineage with low specificity" - this sentence is unclear. Human T cells are usually divided into conventional (abT helpers and cytotoxic cells) and unconventional (MAIT, iNKT and gamma-delta T). What is mentioned above are iNKT (invariant NKT!). Both iNKT and gdT cells are viewed as a link between innate and adaptive immunity - they recognise some specific antigens e.g. sphingolipids for iNKT cells, phosphoantigens for Vd2 gamma delta T or B12 metabolites for MAIT cells. They are capable of rapid response with high amount of cytokines released per cell thus they are responsible for directing the immune response towards Th1/Th2/Th17 pathway.
Authors Response: We thank the reviewer for their comment. The section about immunity has been shortened to be more concise. The section about immune cells has been deleted.
- T follicular helper cells have different abbreviation – Tfh
Authors Response: We thank the reviewer for this correction. The section about immunity has been shortened to be more concise. The sections including these abbreviations has been deleted.
- Lines 145-163 are somewhat misleading. It should be clearly stated that naive B and T cells that recognise antigen presented by APC get activated and undergo further maturation etc. In general the immune introduction should be significantly shortened. This is only a introductory part and should not be 5 pages long. The majority of that part repeats basic immunology for university textbooks - this is completely unnecessary and should be ommitted.
Authors Response: We agree with the reviewer that the section about immunity should be shortened to be more concise. Accordingly, changes have been made to the revised version of the manuscript (Section 2).
- Lines 236-239: this is an oversimplification. It is not only about the activation/suppression balance, but also cytokine balance etc. I would suggest ommitting this sentence.
Authors Response: We thank the reviewer for their comment. We have omitted this sentence and added a new sentence focusing on the role of miRNA in understanding diseases of the immune system: “The immune system is constantly in a dynamic yet delicate balance. MiRNAs may provide insight into understanding diseases of the immune system and possible treatments. This includes autoimmune diseases such as multiple sclerosis (MS) and Rheumatoid Arthritis (RA) [51], as well as allergic reactions [52], and cancers such as leukemia [46].” Lines 147-151.
- Lines 343: while HMVEC are normal endothelial cells from microvasculature, BJAR are Burkitt lymphoma cells. Thus results obtained for BJAB may or may not be relevant to healthy human B cells.
Authors Response: That was a great question and thanks for asking. We agree with the reviewers’ observation. To keep it physiologically relevant, we have deleted the section on BJAB cells and focused on HMVEC-d cells in line 280.
- Lines 346-347: was this during infection (so in vivo) or on cell lines (so in vitro). Those are two separate things. In vitro experiments on cell lines does not replicate the complicated nature of real infection and the network of interactions between virous tissues.
Authors Response: This experiment was performed in vitro. We are aware of the limitations of in vitro experiments and address this in subsection 5.4 Considerations (lines 377-378). Many of these studies are performed in vitro before moving to in vivo models. In addition, we highlight that predicted computer models should also be verified through in vitro and in vivo studies in section 6.3 (lines 477-482).
- "This includes miR-17 and miR-20, which target an HIV protein, as well as miR-28, which targets the HIV transcript, and miR-29 family members, which interfere with HIV replica" - the sole principle of micro RNA is that it targets RNA, not protein.
Authors Response: We thank the reviewer for their comment and have added the following clarification:
“This includes miR-17 and miR-20, which downregulate expression of p300/CBP associated factor (PCAF), an HIV protein cofactor. In addition, miR-28 targets the HIV transcript and miR-29 family members interfere with HIV replication.[15]” (lines 309-312).
- "The p38 MAPK pathway can be activated by cytokines or viral infection." - this pathway is activated downstream of numerous receptors e.g. as part of TCR signalling in T cells.
Authors Response: We thank the reviewer for their comment and have clarified as below:
“The p38 MAPK pathway can be activated by many stimuli, including signaling via cytokines or viral recognition.” (lines 336-337).
- At least part of the mechanisms mentioned in text should be also represented as a figure.
Authors Response: We thank the reviewer for their suggestion and have added Figures 1 and 3 to illustrate the roles of miRNA in immunity and pro-viral vs antiviral effects of miRNA during infection, respectively (pages 5 and 13).
- Simoa is a bead based method to measure various analytes, including direct measurment of some micro RNAs.
Authors Response: We thank the reviewer for pointing out our mistake. We are at fault and accordingly deleted information on Simoa in the revised version of the manuscript.
- Part 8. Discussion should be ommitted - it merely repeats what was said above.
Authors Response: We thank the reviewer for their suggestion and have omitted the discussion. We have relocated some fragments to other sections.
“Since the discovery of miRNA in C. elegans in 1993, and discovery of mammalian miRNA in 2000, the field of miRNA research has grown dramatically.[191] In 2017 alone, there were ~11,000 studies on miRNA, with more than 6,000 of these studies on miRNA diagnostics and therapeutics combined.[20]” has been relocated to the introduction (lines 89-93).
“Population differences in miRNA expression may also inform differences in immunity.[184]” has been relocated to 3. MicroRNA and Immunity (lines 151-152).
“Several miRs have been implicated in regulation of viral infection, however in many cases their exact mechanisms have yet to be elucidated.[118,120] Computational approaches may be used to predict miRNA targets, but can have high false positive rates.[195] Novel algorithms continue to be developed in efforts to improve these predictions.[195-201] Predicting miRNA binding sites can cut down on research costs by narrowing potential targets, however they need to be experimentally validated using in vitro and in vivo models.[195]” (lines 476-482).
and
“In miRNA-based therapeutics, researchers must be wary of off target effects in different cell types, particularly when manipulating levels of miRNA with multiple targets.[172] This is highly relevant when considering possible side effects. For example, miravirsen targets miR-122 in HCV patients and lowers HCV replication but can also lower cholesterol levels.[169] However, less specificity may not be a disadvantage in some cases and may lead to miRNA-based therapies with broad antiviral effects that have the potential to treat multiple viruses.[31,126] Further advances in miRNA therapeutics should aim to improve delivery, stability, tolerability, and specificity for appropriately targeted and prolonged effects that minimize toxicity and do not evoke an immunogenic response.[172,179] In addition, diagnostic and therapeutic research should consider population differences in miRNA expression related to immunity and immune response to viral infections.[184] This may provide insight into how to make therapeutics based on resistant populations or avoid sampling biases in diagnostic devices.[203,204] Nutritional uptake of miRNAs also provides an avenue for immune modulation and may be of interest to researchers investigating therapeutics or possible active ingredients in homeopathic remedies.[182,192]” have been relocated to 6.3. Challenges and Future Considerations (lines 545-560).
- In abstract and conclusions, authors claim that this is a SYSTEMATIC review. There is no sign however. How was it performed e.g. what search querries, databases etc were used? Which studies were included and which were excluded?
Authors Response: We thank the reviewer for their comment. We have omitted the word ‘systematic’ in the abstract and conclusions.

Round 2
Reviewer 3 Report
Bauer et al addressed a few of my concerns, however, I still think fig1 could be drawn in a better way. Secondly, I will reiterate the content is good with limitations and organized well but only a few new things that the community will benefit. The people who are interested in viral microRNA or host microRNA regulating viral outcomes is important to decipher the cleavage versus non-clavage mode of gene regulation. I agree with the authors that only a few cleavage targets are known and this could be only because of less efforts have been made into this. Although direct cleavage targets have been identified two decades before by Bartel and Joshua labs.
Author Response
REVIEWER 3:
Bauer et al addressed a few of my concerns, however, I still think fig1 could be drawn in a better way. Secondly, I will reiterate the content is good with limitations and organized well but only a few new things that the community will benefit. The people who are interested in viral microRNA or host microRNA regulating viral outcomes is important to decipher the cleavage versus non-clavage mode of gene regulation. I agree with the authors that only a few cleavage targets are known and this could be only because of less efforts have been made into this. Although direct cleavage targets have been identified two decades before by Bartel and Joshua labs.
Authors Response: We thank the reviewer for their suggestions. We have revised the graphical abstract to include modes of gene regulation and some big picture effects of miRNA regulation during viral infection. We have also included a new section (2. Modes of MiRNA-Mediated Gene Regulation) to discuss cleavage and non-cleavage methods.

Reviewer 4 Report
All the questions are addressed. I have no further comments and it is now suitable for publication.
Author Response
REVIEWER 4
All the questions are addressed. I have no further comments and it is now suitable for publication. English language fine. No issues detected.
Authors Response: We thank the reviewer for an encouraging remark. We sincerely appreciate it.

Reviewer 5 Report
Authors have acceptably revised the ms.
Some additional suggestions:
1. Interleukins are typically written with dash e.g. IL-1a instead of IL1a (line 131)
2. Line 138-140 - this sentence is too general. Please provide some real, specific insight here.
3. Lines 142-143 - mice deficient in what?!
4. Figure 1 is pointless. It provides no real information...
Language corrections required.
Author Response
REVIEWER 5:
Some additional suggestions:
- Interleukins are typically written with dash e.g. IL-1a instead of IL1a (line 131)
Authors Response: We thank the reviewer for their correction and have implemented it in what is now line 176 in the revised manuscript.
- Line 138-140 - this sentence is too general. Please provide some real, specific insight here.
Authors Response: We thank the reviewer for their suggestion and have added “For example, miR-125b has many targets, including several transcription factors involved in B cell and T cell differentiation.[60] On the other hand, miR-10a targets transcription factors required for monocytopoeisis and megakaryocyte differentiation.[60]” (lines 186-189 of the revised manuscript).
- Lines 142-143 - mice deficient in what?!
Authors Response: We thank the reviewer for their question. We have revised this sentence to “In mice, miR-142 has a role in maintaining cell levels of type 1 Innate Lymphoid Cells (ILCs), NK cell survival, and response to cytokines.[62] This may contribute to the greater susceptibility of the miR-142-deficient mice to Murine Cytomegalovirus (MCMV) infection compared to wild-type counterparts.[62]” in lines 189-192 of the revised manuscript for clarity.
- Figure 1 is pointless. It provides no real information...
Authors Response: We thank the reviewer for their comment; however, this figure was requested by Reviewer 4 as a visual representation of the role of miRNA in immunity. Since they are content with this figure, we believe it would be inconsiderate to remove it now. We have changed the legend to reflect that this is a summary figure in line 205 of the revised manuscript. We hope the reviewer can accept this change.
***To address any further concern on English, we have had the revised manuscript reviewed by Dr. Ron Johnson as well as three other native English speakers (co-authors in this papers). We have also run the article through Grammarly and made suggested corrections.
